# "Face-to-Face Trumps Everything": An Exploration of Tutor Perceptions, Beliefs and Practice Within Blended Learning Environments

**Andrew Youde**

School of Education and Professional Development, The University of Huddersfield,
Huddersfield HD1 3DH, UK; a.youde@hud.ac.uk

**Abstract:** This paper explores the practices of higher education tutors in blended learning contexts. Significantly, the influence of their perceptions on practice was considered by investigating previous teaching and learning experiences, and their views of the affordances blended learning offers adult learners. The analysis was undertaken in relation to these learners' perceptions of their tutors whilst studying part-time, vocationally relevant degrees, at a distance. A mixed methods approach was adopted to conduct a detailed exploration of eight tutors' practice. Data analysis suggested that all tutors had negative experiences of online learning as students with these perceptions appearing to influence their practice. They generally avoided online pedagogies and adopted alternative approaches to their practice, namely a focus on face-to-face delivery with enhanced learner support, which was found to align with their described pedagogical beliefs. These tutors considered online teaching and learning as a deficit in this context.

**Keywords:** blended learning; blended tutoring; online tutoring; tutor perceptions; tutor beliefs

## 1. Introduction

This paper reports findings of a piece of research that explored effective tutors and tutoring within blended learning environments. This context typically involves significant online teaching, learning and support, but includes some face-to-face contact [1] (see Research Context section for further discussions about the nature and structure of the courses under investigation). The research comprised a detailed exploration of eight tutors' teaching, learning, assessment and learner support on one of their modules (see Appendix A for an overview of tutors' key characteristics and roles). These contributed to courses aimed at part-time (PT) learners within higher education (HE), undertaking vocationally relevant degrees whilst, usually, in full-time (FT) employment. There can be difficulties when tutoring these learners, particularly regarding the influence of daily events within their lives, together with the pressures and time constraints of work [2,3]. However, adult learners tend to understand what they want to achieve from education and have clearer goals in mind [4,5]. The research questions were:

- What were tutor perceptions of the affordances and limitations of blended learning?
- How did tutor perceptions of blended learning influence their practice?

Throughout the data analysis process, themes emerged that suggested the influence of tutor perceptions and beliefs on practice when delivering their modules. This paper explores these phenomena, with the analysis suggesting that perceptions and beliefs provided a valuable insight into the actions and motivations of tutors. Further, all tutors had largely taught in face-to-face contexts before moving to some blended learning delivery. This raises notions of technology acceptance [6,7], which, in the context of this research, considers how tutors accept and use educational technologies as part of their module delivery.

This paper defines perceptions to be how teaching practices are regarded and understood by the tutors in this context and, in a similar way to Minor, Onwuegbuzie, Witcher and James [8], the research considers whether these perceptions are related to educational beliefs. It is important to carefully separate a tutor's broader, general belief system from their educational beliefs [9] (p. 316). These educational beliefs are broad and encompass teacher efficacy, epistemological beliefs, tutor self-concept and self-esteem, and self-efficacy. As Pajares summarises:

> The result is a view of belief that speaks to an individual's judgement of the truth or falsity of a proposition, a judgement that can only be inferred from a collective understanding of what human beings say, intend and do. [9] (p. 316).

There is a broad consensus of beliefs being an effectual psychological construct in teacher education [10], notably, for this research, having an impact on practice and a key predictor of teacher behaviour [9]. Research in online learning contexts has also suggested such a link between beliefs and tutor practices [11,12].

Research and accompanying literature exploring tutor perceptions, beliefs and practices within blended learning modules or programmes are both scarce. However, there has been consideration of university tutors' beliefs and practices regarding their uptake of virtual learning environments (VLEs) [13] and web-based technology [14]. Of relevance to this paper, these studies reported that, generally, tutors who were most able to articulate teaching beliefs could justify their use of educational technologies in relation to an overall pedagogical approach [13,14]; however, the influence of post-rationalisation cannot be discounted from their findings. Further, following interviews with six teachers, Lawrence and Lentle-Keenan [14] found that alignment between technology tools and teaching beliefs can lead to a more enthusiastic use of educational technology. Steel [13], in her study of beliefs and VLE learning designs with three experienced technology users, found that tutors were able to align their technology use to their teaching beliefs, but she did not discount that this could have been due to the underpinning strengths of their belief system. The studies cited here, however, placed a stronger emphasis on tutor uptake of educational technologies than this paper intends. Whilst the above limitation of post-rationalisation could be directed at this study, this is partly mitigated by drawing on learners' perceptions of their tutors as part of the analysis.

There is a wide range of traditional and new work roles that distance educators perceive as important to fulfil their jobs successfully [15], which has implications on academic workloads and work practices. Lawrence and Lentle-Keenan [14], found that those who used technology minimally in their general practice cited workload issues, the opposite of which was uncovered for more prolific users. However, their research revealed that all found it difficult to spend sufficient time to design online courses. These sources highlight the potential impact on tutor's perceptions of their workload on their blended learning practice, which is considered as part of this research.

To summarise, research in the area of tutor beliefs in distance education has commonly focussed on technology uptake rather than a holistic examination of module or unit delivery. Furthermore, learner perceptions of their tutors have received limited consideration. This paper seeks to address this gap. This research, particularly, explores teacher beliefs developed in predominantly face-to-face contexts, through both tutors' previous education and teaching, and examines how they have impacted on their blended learning practices.

## 2. Research Context

The research is based at a university in the north of England which has approximately 520 full-time academic staff and 20,000 students. The courses investigated were all within an education disciplinary area. Therefore, the research focussed on this particular subject area, to explore effective blended learning practice through this disciplinary lens. The courses under investigation adopted a day-school model of delivery, where learners typically attend classes one day per month, with the remaining time spent studying independently, utilising resources and communication tools held on

the university's VLE. Modules are usually a term in length (approximately three to four months) from the first day-school until learners submit summative assessments. Each module, therefore, has two or three day-schools, with the overall course structure and delivery models developed by tutors and course leaders in conjunction with course approval committees. This represents approximately 12 to 18 h of face-to-face contact for modules that have an indicative 300 student learning hours. During a module, tutors have responsibility for teaching, assessment and monitoring learner progress. They are required to prepare suitable learning materials for both online contexts and day-schools, but can structure the delivery as they wish.

## 3. Conceptual Framework

For effective teaching and learning in HE, Biggs' [16] Constructive Alignment Model states that all components of teaching and learning are congruent or aligned. Mayes and de Freitas [17] developed this model for online and blended learning contexts. Their three broad theoretical perspectives are:

- The associationist/empiricist perspective (learning as activity);
- The constructivist perspective (learning as achieving understanding through individual or social approaches);
- The situative perspective (learning as social practice).

Mayes and de Freitas [17] (p. 20) state, "most implementations of e-learning will include blended elements that emphasise all three levels: learning as behaviour, learning as the construction of knowledge and meaning, and learning as social practice", and this was found in the modules investigated as part of the research. The constructivist perspective is now outlined with discussion of its alignment, particularly with regard to teaching, learning and assessment in blended and online contexts. This perspective showed most congruence with the practices found on the modules investigated.

The constructivist perspective has both an individual and social focus, to allow learning as achieving understanding in both individual and collaborative contexts [17]. Fox [18] (p. 31) argues that it is important for tutors to realise that students are always trying to make sense of their study in terms of what they already know. This was relevant for the learners in the research study who were trying to apply their learning to work practices. The learners were professionals and brought a breadth of existing knowledge, understanding and experiences to the classroom [18] (p. 29), and this highlights the relevance of analysing the data in relation to constructivist learning theory.

The individual constructivist perspective highlights the achievement of understanding through active discovery, where learners construct new ideas by hypothesis testing. The pedagogy aligning with this perspective includes interactive environments for knowledge expansion, cognitive scaffolding, experimentation with the discovery of principles, adaptation of teaching to existing student understanding, and support for reflection, analysis and evaluation. Assessment strategies aligning with this perspective encourage experiential learning, experimental learning, problem-based learning, case-based learning and self-evaluation, and autonomy in learning. When evaluating tutor practices as part of the research, I considered that this perspective primarily focussed on students generally learning independently from tutors and peers throughout modules. However, tutors provided support to learners and engaged in dialogue regarding learning and assessment, but with limited peer interaction and collaboration occurring outside face-to-face contexts.

The social constructivist perspective highlights the achievement of understanding through collaboration and dialogue. The pedagogy aligning with this perspective includes interactive and collaborative environments leading to conceptual development; support for reflection, peer review and evaluation; and experimentation with shared discovery. Assessments aligning with this perspective are common to the individual constructivist perspective, however, include collaborative activities, participation, peer review and shared responsibility. When evaluating tutor practices as part of the research, I considered that this perspective included a far greater focus on peer collaboration throughout teaching, learning and assessment, particularly outside face-to-face contexts and within online learning environments, such as the VLE.

Mayes and de Freitas' perspectives provide a framework to evaluate practices across a unit of study. This research undertakes a holistic examination of a tutor's approach to the delivery of a module, therefore, the framework is relevant to underpin the analysis of practices in this context.

## 4. Methodology

An explanatory mixed methods design [19] (p. 81) was adopted, to conduct a detailed exploration of tutors' practice on one of their modules. This firstly involved issuing a questionnaire to ascertain learners' perceptions of tutors, and the teaching, learning and assessment they experienced whilst studying the module. A qualitative analysis followed, via tutor interviews and VLE content analysis, which explored approaches to teaching and learning appearing, to influence learner perceptions. The tutor sampling criteria applied were:

- they delivered the module on a 'day school' basis, that is, where learners attend university for a small number of days (typically two or three), with remaining teaching conducted via computer mediated communications (CMCs);
- they were an experienced teacher/lecturer (over five years), and had delivered at least three previous modules in blended learning contexts;
- their learners were studying undergraduate or post-graduate courses on a part-time basis;
- their learners were studying qualifications relevant to their profession.

Participant privacy was supported by selecting pseudonyms for the tutors, based on popular UK names.

A random selection of students (n = 72, 64% response rate, covering the eight modules investigated) completed the questionnaire, which was designed to elicit the general opinion about the quality of tutoring they experienced. To obtain this, a modified version of the Course Experience Questionnaire (CEQ) was used [20]. The original questionnaire was designed as an indicator of teacher effectiveness on courses in HE institutions and draws on learners' perceptions of teaching, curriculum and assessment. It was originally designed for courses with traditional approaches to teaching, with a more regular tutor/learner contact than day school models of delivery allow. It was modified to make it suitable for an individual tutor (see [21] for a similar use of the CEQ), and a blended teaching model (see [22] and [23] for a similar use of the CEQ in distance education). The scale items adopted for this research were largely the same as the original CEQ, but adapted in line with Kreber's [21], Richardson and Woodley's [22] and Richardson's [23] studies, and were:

- good teaching communication;
- good teaching feedback on, and concern for, student learning;
- clear goals and standards;
- appropriate workload.

Descriptive statistics generated from the questionnaire provided a broad overview of learner perceptions and a ranking of tutors, which then allowed the qualitative data to explain and build upon the initial quantitative results.

Qualitative data were gathered from interviews with tutors, which explored their approaches to delivery and factors that impacted on quality, and an analysis of the content and communications in the VLE, which provided insight into their online practice. Tutor interviews were approximately an hour long and explored their backgrounds and previous roles; their approach to teaching, learning and assessment during the module; resource and workload implications whilst delivering the module; and tutor feedback on their students. Template analysis was chosen to analyse both the tutor interview data and VLE communications [24]. King [24] (p. 256) argues that template analysis is not a single method or research itself, or a methodological position, but a series of techniques for the inductive analysis of textual data. The first stage of the qualitative analysis was to carefully listen to each interview. Notes were made about what appeared to be relevant themes worthy of further investigation; however, the real benefit of this process was to listen to the manner in which points were made. It was illuminating to note expressions of confidence, concern, anxiety and enthusiasm, and was particularly helpful when trying to analyse tutor perceptions and beliefs. The first template had a mix of descriptive codes, such as tutor experience, and analytical codes, for example, tutor

ability to work within available resources. Flexibility was required in template development and subsequent analysis, particularly as themes were developing around tutor perceptions and beliefs appearing to align with learner feedback. Themes, such as tutor self-efficacy, were noted as the coding process was undertaken and were analysed using a framework approach to thematic analysis [25] (p. 550), which involved tabulating emerging ideas against tutors (who were ranked in descending order of learner perceptions, measured by CEQ scores). Care was taken to explore the relationships between themes beyond the linear structure of the template, such as tutor perceptions of online teaching and learning, which allowed analysis across various strands of the research. Through this process, themes emerged that were important in all modules; important in those of tutors receiving the highest CEQ scores; and those that were only observable in the tutors receiving lower scores.

Data collection for this article occurred between March and July, 2011, however, the student groups investigated are representative of cohorts found within education disciplines, generally, today. As stated earlier, there is limited research into the influence of tutor perceptions and beliefs in blended learning context, and, given that there were over half a million part-time students in the United Kingdom during the 2018/19 academic year [26], research for this group is still relevant.

## 5. Quantitative Data Findings

The CEQ provided feedback on tutor effectiveness on the modules and measured learner perceptions of teaching and assessment. Learners gave answers to five-point Likert scale questions, with the results detailed in Table 1.

**Table 1.** Tutor's Course Experience Questionnaire (CEQ) Results—Mean and (Standard Deviation).

| Tutor (Pseudonym) | N | CEQ Total | Clear Goals and Standards | Good Teaching Communication | Good Teaching Feedback | Appropriate Workload |
|---|---|---|---|---|---|---|
| Ann | 6 | 4.06 (0.36) | 4.29 (0.40) | 4.50 (0.35) | 4.25 (0.45) | 3.71 (0.68) |
| Bill | 7 | 3.86 (0.64) | 3.96 (1.09) | 4.33 (0.69) | 3.93 (0.72) | 3.89 (0.66) |
| Claire | 7 | 4.10 (0.35) | 4.29 (0.57) | 4.62 (0.36) | 3.64 (0.35) | 4.25 (0.64) |
| Daisy | 4 | 3.23 (0.45) | 3.69 (0.24) | 3.58 (0.74) | 2.87 (1.18) | 2.94 (0.31) |
| Emily | 15 | 3.99 (0.26) | 4.42 (0.28) | 4.15 (0.44) | 4.12 (0.44) | 3.48 (0.47) |
| Frank | 5 | 3.43 (0.37) | 3.45 (0.62) | 4.00 (0.97) | 3.30 (0.67) | 2.90 (0.88) |
| George | 14 | 3.55 (0.40) | 3.59 (0.74) | 4.31 (0.53) | 3.68 (0.56) | 2.99 (0.55) |
| Harry | 14 | 3.42 (0.67) | 3.42 (1.15) | 3.55 (1.06) | 3.13 (0.80) | 3.46 (0.47) |
| Mean | | 3.72 (0.53) | 3.89 (0.83) | 4.12 (0.79) | 3.73 (0.77) | 3.38 (0.63) |

Preliminary analysis revealed a CEQ mean total of 3.72, and similar high scores were evident across the constituent scales, indicating that learners considered their tutors to be effective. Standard deviation scores were relatively small for a five-point scale, which suggested a common perception from the groups of learners. Overall module pass rates, whilst a crude measure of educational success, were greater than 95%, with some of the remaining 5% expected to complete in the near future. Learners were asked to rate their module achievement on a five-point scale (very disappointed to very good) and the resultant mean score was 3.83, indicating broad satisfaction with their results and academic development. During the interviews, when tutors were asked to provide an overall impression of their groups, there was a consensus around motivated learners, engaged in their study,

and producing good quality work. These indicators, together with the generally high CEQ scores received, suggested successful modules, with students learning and achieving.

Tutors Ann, Bill, Claire and Emily received the higher CEQ scores, and, generally, their approaches involved limited learner-peer interactions. Whereas, Daisy, Frank, George and Harry, receiving lower scores, tried to adopt greater social constructivist approaches. Whilst there were other influencing factors to CEQ scores, including, as Fives and Gill [10] noted, tutor beliefs being an effectual psychological construct, the following section explores this important finding.

## 6. Findings from the Interviews and VLE Analysis

### *6.1. Introduction*

The following sections were discerned from the qualitative analysis and represent a selective approach to identify the themes that were most relevant due to their influence on learners [24]. A significant finding from this research was the lack of tutor-peer and peer-to-peer interaction within the online elements of all modules. This meant that there was a lack of text to analyse within the VLE, however, this finding did provide great insight into the influence of tutors' perceptions and beliefs on their practice.

### *6.2. Tutor Perceptions of Blended and Online Learning*

The Research Context section highlighted the autonomy tutors had on modules, thus allowing them to develop their own practices within blended learning contexts. This section provides evidence to indicate that their pedagogic beliefs and previous experiences of online learning were influencing the adopted approaches.

All tutors' perceptions of purely online learning were negative, and this was mainly rooted in previous negative learning experiences in this context. Each tutor had some experience as a student in a purely online or distance context. Most found the experience isolating, and talked about feelings of disorientation. When studying, tutors outlined limited engagement with peers in online environments, with reasons given including time, superficial discussions, contributions a "tick-box exercise" (George), and a lack of trust that would have developed in face-to-face meetings. Further, four tutors (Ann, Claire, Daisy, Frank) found the online elements impersonal and lacking human contact. The implications of these perceptions on practice are considered throughout this section. However, it is interesting to note overlaps of these perceptions, and tutor comments about limited learner collaboration within online elements of their modules.

Tutors described three broad reasons for the lack of engagement and learner collaboration within online environments, namely negative student experiences on previous modules; VLE access and user issues; and time for tutors to develop and manage online activities. Ann and George noted difficulties in encouraging engagement when this had not been required or encouraged in previous modules, and reported learners' feelings of frustration at the prescribed nature of collaboration. Ann stated:

> This is the last module of a three-year programme, and they hadn't engaged particularly well with discussions online and I don't think that would have been a time to start with that... When asked about the Blackboard [University VLE] element, a lot of them said they didn't like it, they were not comfortable with online. (Ann)

Learners informed Emily and Harry that the VLE was "clunky", slow and impersonal, and reported access issues. These issues were often found to be user error when investigated, with Harry noting "it's just they haven't got that confidence to give it a whirl or they're looking in the wrong module", but this point did overlap with the third issue of time, which was a concern for three tutors.

The day school model was described to be pressured, with Harry stating that there was limited time to orientate learners around the VLE content and tools. He outlined that some learners had not developed effective use of the VLE in course inductions. This was a barrier to learning and a

particular problem, as his learners were still in the first year of study within HE. Further, Daisy and George, who received lower CEQ scores, stated that they did not have sufficient time to effectively set up and manage pedagogically appropriate opportunities for online engagement and collaboration. Daisy commented:

> I think I could have done more. I have tried over the year, bits with different
> groups and now because I think it is a time issue as well, I don't have time
> to change everything and also with having a few new modules to teach this
> year.

This quote suggests tensions between a preference for social constructivist pedagogy within Daisy's teaching and the challenges experienced of developing activities aligning with this approach.

To review tutor perceptions of online learning further, their responses to questioning about the extent of engagement and collaboration provided insights. When tutors were questioned about strategies to motivate, support learning, build and maintain relationships, and encourage pedagogically appropriate learner collaboration in online environments, the most common response was to avoid the question. Tutors were generally confident in their approaches to teaching the modules. However, they provided limited responses about online teaching, engagement and learning. There was no discernible pattern to answers in relation to learner feedback, with those receiving higher CEQ scores giving limited responses and even resorting to saying "be available", "point them in the right direction", and "monitor online activities". Generally, tutors avoided the question by outlining their practice in face-to-face environments, which was revealing of their confidence in purely online contexts. Tutors shifted the discussion to areas where they were comfortable, predominantly face-to-face teaching and learner support, as these appear closer to their pedagogic beliefs about effective practice. When considering notions of technology acceptance [6,7], this suggests tutors are not accepting and using the affordances of the VLE to communicate with students or encourage peer-to-peer interaction. Issues of tutor efficacy are explored later (see Tutor self-efficacy section), however, tutor perceptions were more positive towards the affordances of blended learning.

Although all tutors expressed some scepticism regarding blended learning and the day school model of delivery, they were generally more appreciative of the affordances offered, particularly for the type of learner investigated within this research study. Emily commented that "face-to-face trumps everything", and this summed up the sentiment of a group of lecturers more accustomed to traditional approaches to teaching. Daisy's comment was also illustrative of tutor descriptions of practice, "we tended on the day schools to talk and discover, sort of deconstruct sometimes, concepts and issues that are good for discussion". She continued to note the difficulties of learners discussing sensitive issues in online environments during her Equality and Diversity module, preferring these to occur at day schools. Tutors outlined that day schools offered increased opportunities, which included richer social constructivist approaches, instruction and clarification of assessment requirements, motivational opportunities, and the development and maintenance of relationships. Bill and Ann also spoke about the time that day school models provide, particularly for learner reflection.

Tutors receiving higher CEQ scores were noticeably more positive about blended learning and the day school model of delivery, particularly focusing on the affordances of face-to-face elements and suitability for this type of learner. Emily stated, "you want them to feel like they want to participate by being positively motivated by the course and I think the day schools are quite critical in raising motivation. They do feel motivated particularly at the day schools".

Tutors receiving lower CEQ scores made more negative comments, particularly around the time available for face-to-face contact, which indicated a greater sense of frustration towards this teaching model. Four tutors expressed concern about time, with Frank commenting that this form of teaching allowed "delivery of the basics in the time available", and this summed up perceptions. Frank developed this point; "so, for example, 10 years ago the delivery pattern was weekly, in class sessions over an academic year. 24 weeks of 2 h classes in the evening. That got knocked back and now it's just two Saturdays". An influencing factor here was that Frank and Daisy's modules had previously

been taught weekly and they stated that this restricted their ability to deliver the module in the time allowed. Arguably, such feelings of frustration are likely to influence learners' perceptions of the module if tutors are making reference, albeit subtly, to the time available for delivery.

*6.3. Tutor Perceptions of their Workload and its impact on Practice*

Data analysis suggested that tutor workload was influencing learner perceptions of modules, however, all stated that any workload issues did not affect the overall learner experience. Further, tutors advised that they were happy with response times to queries and the return of formative and summative feedback. These factors appear relevant to the general module success, given the high scores received via learner questionnaires and, in particular, the CEQ scale items 'good teaching feedback' and 'good teaching communication'.

An analysis of perceptions of workload and tutor's CEQ scores revealed some patterns, but also some anomalies. Ann and Claire, receiving higher CEQ scores, made less reference to workload issues and outlined clearer approaches to teaching, learning and assessment. They described availability and active support, however, as this was most evident through e-mail and not the VLE, it was difficult to validate. Emily, who received one of the higher CEQ scores, did outline a number of workload issues, including such comments as "feedback is so time consuming" and "Twitter saps energy", but was adamant that she was available and response times were good. Further, comments were made about improvements to the module that could be made given enough time; nevertheless, she felt "it worked so there was no urgency". This issue of improvements was raised by Daisy, Frank and George who, again, cited time and general module success as reasons for not initiating change. Tutors receiving the lower CEQ scores expressed greater frustrations with workload issues and identified more possible module improvements, time permitting. To illustrate, Harry found teaching both traditional face-to-face and blended learning courses concurrently to be problematic, and stated, "I have found it really difficult to balance both approaches at the same time".

This research only considered tutor and learner perceptions, and did not compare the impact of actual workloads on practice; therefore, the comments outlined above could be justified. Daisy, for example, worked part-time, which could explain the relatively high number of comments made about workload. However, tutors' feelings and frustrations around workload could potentially be influencing learner perceptions of module quality. If learners feel that their tutor is busy, they may be discouraged from engaging in a dialogue. Consequently, tutor perceptions of workload appear to be a relevant factor when exploring effective blended tutoring.

Workload issues also arose in the following two sections as tutors' reflections on practice, and self-efficacy in blended learning contexts, are discussed.

*6.4. Tutor Reflections on Practice*

Ann and Claire, who received the highest CEQ scores, were forthright and confident in their approach to the module, with reflections focusing on the lack of support they had received when studying at a distance. In addition, both had considerable pastoral care experience in previous roles at colleges of further education (FE). Their reflections had resulted in an approach that was clear, structured, pragmatic, and student-centred, with a strong emphasis on available support. Both outlined difficulties found whilst studying at a distance, particularly regarding the amount of support received and opportunities for dialogue with lecturers, and their respective actions help understand their approach to module delivery. Ann said that her solution to a lack of support was to "just get on with it", a phrase used four times within the interview. Throughout her studies, she gave up on trying to communicate with lecturers and, back then, discussions with peers largely occurred in class. Therefore, an independent and instrumental approach to her studies was described, that used module specifications, which included the intended learning outcomes, assessment briefs (although she stated that these were sparse in detail), and module syllabi. She described using all this information to produce, what she felt, was required for the assignments. Reflecting on this experience led her to include extensive support for assessment and be available to learners. Ann and Claire's modules were developed to focus on assessment, with both stating that they were proactive in

promoting dialogue and being available for support. However, this was commonly provided within e-mail and face-to-face communications.

As stated in the Introduction, research has noted that tutors who are most able to articulate teaching beliefs could justify their use of educational technologies in relation to an overall pedagogical approach [13,14]. Ann and Claire's descriptions of practice provide evidence to support this position. Furthermore, whilst Steel [13] found that experienced technology users were able to align technology use to their beliefs, it appears that Ann and Claire, who were experienced in face-to-face contexts, could also achieve such alignment.

When tutors were asked to consider improvements to practice during modules, two responses emerged, and one of these was surprising. Firstly, Emily and George stated that there was too much content on the VLE, which needed simplifying and more structure. Secondly, Ann, Bill and Harry stated that they should be engaging learners more in online environments and encouraging greater dialogue and peer collaboration. Ann described "guilt", as she outlined that there was insufficient online teaching, engagement and learning, which was surprising, given the confidence demonstrated in her approach to module delivery. She had acknowledged the success of learners, was pleased with the quality of work received, and was happy with module feedback. Ann's view illustrated the challenge to tutors' pedagogical beliefs provided by the day school model for those more accustomed to face-to-face delivery. The desire for greater online social constructivist pedagogy was even more surprising, given the negative comments already outlined by tutors when they undertook online learning as students. These tutors appeared to feel guilt over the lack of social constructivist learning even though modules were generally successful, and they had not engaged in such activities when studying, either purely online or at a distance.

Tutor reflections about their modules develop the discussion from the previous section regarding their perceptions of workloads. Daisy and George, who received lower CEQ scores, reflected on a greater number of difficulties of teaching and learning in this context, and noted improvements they would like to make. Daisy was the least experienced tutor under investigation, and still adjusting her practice from face-to-face to blended learning contexts. Additionally, this was the first iteration of the module being taught on a day school basis and she reflected on the difficulties, particularly around time, of adjusting to a new delivery model. She commented:

> I think the tensions are with the time and part of it is building your own
> knowledge as well and at the moment, I am at a level where there are a lot
> of new things to discover so it's not like I know a lot already.

However, George had taught the module four times and had not made improvements that he felt were needed. He commented "it lends itself to a different assessment to be honest", and whilst workload was cited as the key reason behind this, there appeared to be self-efficacy issues, particularly around module management. Further comments suggested this, including "which maybe, in hindsight, would work better" and "some of that could be the way I am selling it", when discussing improvements to practice.

Emily's and Harry's reflections centred on issues of workload and managing competing pressures of effective pedagogy and learner support in blended learning contexts, with other demands placed on their time. The two tutors received differing CEQ scores, with their levels of learner support described appearing significant. Whilst Emily, the tutor receiving the higher score, talked of the difficulties of competing pressures, she was adamant that response times were quick, feedback was prompt, and was active in supporting learners between day schools. Harry, who received a lower score, was also adamant that workload issues did not impact on the student experience. However, he gave no examples of proactive support. This comment illustrates Harry's reflections; "I think the nature of these courses is people come in for the two days of each module and if they want to disappear for a month and just get on with their work, they can". It describes his trust in learners and hints that support is available if needed, but indicates a potential lack of proactive support.

The analysis of tutor perceptions and reflections has unearthed issues about tutor self-efficacy, both in face-to-face and blended learning contexts, which will now be discussed in more depth.

*6.5. Tutor Self-Efficacy*

According to Bandura [27] (p. 2), self-efficacy is "the belief in one's capabilities to organize and execute the courses of action required to manage prospective situations". Furthermore, self-efficacy in tutors can improve performance, as Bandura [28] (p. 196) continues, "strong self-efficaciousness intensifies and sustains the effort needed for optimal performance, which is difficult to achieve if one is plagued by self-doubts". The analysis in previous sections could suggest that tutors had self-efficacy in the face-to-face elements of delivery, but not the online contexts.

A common theme among tutors was the positive comments and enthusiasm shown about teaching in face-to-face contexts, indicating self-efficacy and, therefore, potentially a factor in the success of the modules. A flavour of the comments includes:

> I think my teaching style is quite an enjoyable one. I enjoy it! (Ann);
>
> They do feel motivated, particularly at the day schools (Emily);
>
> You get them in a good mood and they're excited to be there (Harry).

The tutors were enthusiastic and believed in their capabilities at day schools to teach and motivate learners to succeed. With the majority of teaching occurring at day schools, tutors' self-efficacy in this context appears an important aspect in the high CEQ scores received.

Tutors' self-efficacy in online environments was generally limited, resulting in differing actions. Ann, Claire and Frank were confident in their answers that CMCs were used for support, and not teaching and learning. Three tutors used either wikis or discussion boards with mixed success, and comments mirrored this when analysed for self-efficacy. George's comment was illustrative, "there is probably less blended learning going on there for a number of reasons, some of which are down to me". Daisy, however, made the most negative comments regarding her approach to the module, which indicated limited self-efficacy. An early comment in the interview set the tone; "it might be more in terms of what I intended to do rather what I actually do". Although she used an instructive approach to teaching key equality and diversity legislation, subsequent examples of teaching practice discussed were social constructivist in orientation, such as, "working in groups and bringing back the information together". She found adjusting the module to blended learning difficult, in particular, with regard to the time for delivery, and it was clear the approach did not align with her apparent social constructivist pedagogical beliefs. "I know that's the way it has to be done so I have to make it work", was a comment that illustrated tensions the approach was causing. Daisy was confident in face-to-face contexts, however her limited self-efficacy in online contexts, and with blended learning approaches in general, could have had a negative influence on learner perceptions of quality during the module.

Harry provided an interesting perspective by describing high self-efficacy in all contexts, whilst receiving the second lowest CEQ score. He spoke confidently about supporting and coaching other tutors with regard to technology and pedagogy in online contexts. Bandura [28] (p. 196) argues that those who perceive themselves as highly self-efficacious feel they need to invest little effort in the achievement of outcomes. This could have been a consideration, as he was not proactive in supporting learners and did not contribute to the module discussion boards.

## 7. Conclusion

This paper has highlighted the influence of tutor perceptions and beliefs on practice within blended learning courses. It has provided a holistic examination of eight tutors' approach to module delivery. Factors associated with general module success have been identified, however, there are observable trends in the data in relation to tutor's CEQ scores, with these appearing to influence learner perceptions of quality. A significant finding from this research was that those tutors who received higher CEQ scores generally adopted an individual constructivist approach to module delivery, whereas those who tried to include greater social constructivist approaches received lower CEQ scores. There appeared to be several issues underpinning this finding.

All tutors had negative experiences of online learning when studying. These perceptions appeared to influence practice on modules and were helpful in understanding adopted approaches. Online teaching and learning were considered as deficits, by these particular tutors, in this context. The quote "face-to-face trumps everything" (Emily) captured the sentiment of a group of tutors more accustomed to traditional approaches to teaching. They adopted alternative approaches other than online teaching, engagement and learning, which aligned with pedagogical beliefs, namely face-to-face delivery enhanced with learner support. This provided an interesting finding when considering notions of technology acceptance [6,7], as it suggested that tutors were not drawing on the affordances of the VLE when planning and teaching in this context. Interestingly, although tutors were generally confident in their approach to modules, a lack of online learning was perceived as bad practice by some, potentially in conflict with their preferred socially constructivist pedagogical approaches and beliefs. Of particular note, some tutors appeared to feel guilt over the lack of social constructivist learning on their modules, even though they were generally successful, and they had not engaged in such activities, as students, in online contexts. The analysis of tutors' responses suggested self-efficacy was important in influencing practices on modules.

Tutor perceptions of blended learning were more positive, in part, due to the face-to-face component. Further, some suggested that it allowed learners to balance study with work and family commitments. There were noticeable trends from learners' CEQ feedback, as tutors' perceptions revealed interesting responses to delivery in blended learning contexts. Tutors receiving lower CEQ scores, on the whole, adopted a 'blame' response, predominantly around time affordances, but also about limited opportunities for social constructivist pedagogy. Tutors receiving higher CEQ scores outlined greater opportunities afforded by the delivery model, including learner support and increased space for reflection and learning. This suggests that tutors who perceive blended learning as an opportunity to enhance practice and meet adult learner needs are considered more effective by their students.

Further significant findings included tutor perceptions of workload, module design and the influence of positive perceptions. Tutors achieving higher CEQ scores generally perceived their workload as manageable and outlined clearer approaches to module delivery. Modules that were originally designed for blended learning contexts were rated better than those adapted from weekly delivery models. Furthermore, this influenced tutor perceptions, particularly around time for delivery, potentially shaping learners' feedback on the module. Generally, tutors achieving higher CEQ scores were more positive about their modules, again, potentially influencing learners' perceptions.

The findings suggest a number of criteria for the recruitment of tutors for blended learning courses. They should have self-efficacy in online, blended and face-to-face contexts, whilst understanding the needs of adult learners. They should be motivated, evidenced through meeting self-esteem needs through, for example, a strong commitment to supporting and developing learners. Further, a clear approach to teaching, learning and assessment that is aligned to a blended learning design perspective, such as individual constructivism, is needed. Careful consideration should be given to tutors showing high self-efficacy, as they may invest little effort in the achievement of outcomes.

Whilst this paper has raised interesting findings regarding tutor perceptions and beliefs, and their actions on a blended learning module, there are some limitations to this research that need to be acknowledged. Firstly, it was located within an education disciplinary area and the findings may have differed if a broader mix of disciplines were included in the study. Further research is needed across a range of subject areas and HE institutions. Secondly, the research was largely based on CEQ scores as the external indicator of outcomes and, therefore, employed a relatively crude measure of blended learning effectiveness. Any future studies could undertake qualitative research of learners' experiences on a module, with this, perhaps, presenting a different understanding of this area. Finally, the findings were potentially limited by the lack of online communication found in module VLEs, although this was revealing of tutor perceptions and beliefs on their practices. This could have been influenced by notions of technology acceptance [6,7], or some tutors' views that they had

insufficient time to prepare and deliver blended learning modules. This, again, should be more rigorously investigated in any future research in this area.

**Funding:** This research received no external funding.

**Acknowledgments:** I wish to thank Kevin Orr and Wayne Bailey for their helpful comments on an earlier version of this article.

**Conflicts of Interest:** The author declares no conflict of interest.

## Appendix A

**Table A1.** Tutors' key characteristics and roles.

| Tutor | Tutor Characteristics and Role |
|---|---|
| Ann | Female, 61 years old. Senior Lecturer and Course Leader for a MA Education programme. Lectures on a variety of HE courses at both a large Further Education College and within the University. |
| Bill | Male, 47 years old. Senior Lecturer in Post Compulsory Education. Has course management responsibilities (BA Education and Training). Lectures on a variety of courses within the School of Education. |
| Claire | Female, 52 years old. Senior Lecturer and Course Leader for the BA Educational Management and Administration. Lectures on a variety of courses within the School of Education. |
| Daisy | Female, 50 years old. Works part-time. Senior Lecturer in Education. Has course management responsibilities (BA Education and Professional Development). Lectures on a variety of courses within the School of Education. |
| Emily | Female, 48 years old. Senior Lecturer and Course Leader for a PT MSc (Multimedia and Education). Lectures on a variety of courses within the School of Education. |
| Frank | Male, 43 years old. Senior Lecturer in Education and Training and Course Leader for a PT Certificate in Education. Lectures on a variety of courses within the School of Education. |
| George | Male, 39 years old. Senior Lecturer in Education.　Lectures on a variety of courses within the School of Education. |
| Harry | Male, 35 years old. Senior Lecturer in Early Years with course management responsibilities (BA Early Years). Lectures on a variety of courses within the School of Education. |

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
