# Peer review of "“Face-to-Face Trumps Everything”: An Exploration of Tutor Perceptions, Beliefs and Practice Within Blended Learning Environments"

_education, doi:10.3390/educsci10050147_

Round 1

Reviewer 1 Report

RESEARCH CONTEXT

There is a body of literature on the relationship between tutors' subject specialism, pedagogical orientation and factors such as gender and length of service. However, Lines 81-2 say only that the tutors were from "an education disciplinary area" – but this could encompass a range of Education specialists from visual artists to educational technologists. More detail is needed of the tutors' backgrounds.

CONCEPTUAL FRAMEWORK

The literature review concentrates on blended learning and (generic) teachers' beliefs and would have benefited from discussion of the literature on teacher acceptance models for ICT (such as the Technology Acceptance Model and Unified Theory of Acceptance And Use of Technology).

METHODOLOGY

A weakness of the research model is its reliance on CEQ scores as the only external indicator of outcomes. This limitation should be acknowledged in the Conclusion.

FINDINGS

Regarding Table 1, the use of means and standard deviations for CEQ totals is largely meaningless, as these have been aggregated from four equally weighted, but very different, subcategories. However, I think this is a general point about the futility of all (managerialist) attempts to quantify the subtleties "student experience" through the blunt instrument of questionnaires.

A weakness of the study is the lack of detail in its qualitative data collection and analysis. It is not evident how and for how long the interviews were conducted. It is also not evident how the interview data were analysed: was, for example, emergent coding employed and what themes were identified? What criteria were employed in the VLE analysis and why are the outcomes not presented?

Some tutors' criticisms [Lines 296-305] appeared to be about the time allocated to course delivery as well as to the blended learning mode. Perhaps this could be followed up in subsequent studies.

CONCLUSION

Line 449: "influencing" should be "influence"

Lines 449-452. The sentences: "A significant finding from this research was those tutors receiving higher CEQ scores generally adopted an individual constructivist approach to module delivery. Whereas, those who tried to include greater social constructivist approaches received lower CEQ scores." might be more clearly expressed as: "A significant finding from this research was that those tutors who received higher CEQ scores generally adopted an individual constructivist approach to module delivery, whereas those who tried to include greater social constructivist approaches received lower CEQ scores."

The final paragraph [Lines 483-490] is rather confused and should be redrafted. Essentially, as it's about criteria for the recruitment of tutors for blended learning courses, the topic sentence should be something like: "The findings suggest a number of criteria for the recruitment of tutors for blended learning courses."

OVERALL

This study focuses on a small group of Education tutors (some part-time) and adult part-time learners. However, while not typical of mainstream undergraduate HE practice it may still be significant in providing evidence that challenges upbeat nostrums around the benefits of blended learning. In this respect it is also original. In terms of rigour, it is lacking (see comments on FINDINGS).

The study does deserves follow-up work: perhaps with a range of tutors in a range of academic disciplines in which blended learning is a significant part of course delivery. Any follow-up studies should also employ richer evidence of educational effectiveness than student questionnaires and tutor interviews (for example, student interviews and/or focus group discussions).

Author Response

Many thanks for taking the time to provide this detailed review of my article and I appreciate the feedback.  My response to each point is in bold below.  Andy. 

RESEARCH CONTEXT

There is a body of literature on the relationship between tutors' subject specialism, pedagogical orientation and factors such as gender and length of service. However, Lines 81-2 say only that the tutors were from "an education disciplinary area" – but this could encompass a range of Education specialists from visual artists to educational technologists. More detail is needed of the tutors' backgrounds.

Response: An appendix has been added (line 535) that outlines key tutor characteristics (gender, age) and their roles and backgrounds.  This signpost to the appendix is at the first mention of the tutors (line 25) and not later in the Research Context section. 

CONCEPTUAL FRAMEWORK

The literature review concentrates on blended learning and (generic) teachers' beliefs and would have benefited from discussion of the literature on teacher acceptance models for ICT (such as the Technology Acceptance Model and Unified Theory of Acceptance And Use of Technology).

Response: Reference has been made to technology acceptance in general as part of tutors’ move from teaching in face-to-face to a blended context.  The behavioural elements of these models are relevant for this paper and brief reference to this body of work has been made in the findings (lines 294-296) and conclusion (lines 483-485 & 527-529).  The following has been added to the literature review (lines 37-40):

Further, all tutors had largely taught in face-to-face contexts before delivering some of their modules in a blended learning context.  This raises notions of technology acceptance [7-8], which, in this context, considers how tutors accept and use educational technologies as part of their module.

METHODOLOGY

A weakness of the research model is its reliance on CEQ scores as the only external indicator of outcomes. This limitation should be acknowledged in the Conclusion.

Response: this weakness, along with the comment regarding ‘employ richer evidence of educational effectiveness’, have been added to a paragraph about the study’s limitations at the end of the conclusion (lines 522-526). 

FINDINGS

Regarding Table 1, the use of means and standard deviations for CEQ totals is largely meaningless, as these have been aggregated from four equally weighted, but very different, subcategories. However, I think this is a general point about the futility of all (managerialist) attempts to quantify the subtleties "student experience" through the blunt instrument of questionnaires.

Response: this limitation has been acknowledge in the conclusion (see above comment - lines 522-526).  The quantitative element of this paper was relatively short with the CEQ providing (as stated in the paper) “a ranking of tutors, which then allowed the qualitative data to explain and build upon the initial quantitative results”.  The qualitative analysis is a far greater focus of the paper and, therefore, provides some mitigation against this concern. 

A weakness of the study is the lack of detail in its qualitative data collection and analysis. It is not evident how and for how long the interviews were conducted. It is also not evident how the interview data were analysed: was, for example, emergent coding employed and what themes were identified? What criteria were employed in the VLE analysis and why are the outcomes not presented?

Response:  Added to the methodology section regarding the interview length and more detail about their structure (lines 181-184).  Template analysis was used to analyse the data – examples of the themes have been included in the methodology section that align with this approach to data analysis (lines 195, 200).  More detail regarding the process of data analysis has been added particularly around the value of listening to the interviews for this research study (lines 187-191) and avoiding a ‘linear’ approach to data analysis (lines 198-201).  Regarding the comment about VLE analysis, a key finding from the research was the lack of interaction within the VLE.  The following has been added to the start of the qualitative data findings section make this point more clear (lines 234-238): 

A significant finding from this research was the lack of tutor to peer and peer-to-peer interaction within the online elements of all modules.  This meant there was a lack of text to analyse within the VLE, however, this finding did provide great insight into the influence of tutors’ perceptions and beliefs on their practice.

Some tutors' criticisms [Lines 296-305] appeared to be about the time allocated to course delivery as well as to the blended learning mode. Perhaps this could be followed up in subsequent studies.

Response:  This point has been added in the final paragraph about the paper’s limitations (lines 527-529). 

CONCLUSION

Line 449: "influencing" should be "influence"

Response: this has been changed (line 470), thank you. 

Lines 449-452. The sentences: "A significant finding from this research was those tutors receiving higher CEQ scores generally adopted an individual constructivist approach to module delivery. Whereas, those who tried to include greater social constructivist approaches received lower CEQ scores." might be more clearly expressed as: "A significant finding from this research was that those tutors who received higher CEQ scores generally adopted an individual constructivist approach to module delivery, whereas those who tried to include greater social constructivist approaches received lower CEQ scores."

Response: this has been changed (lines 470-473), thank you – it now reads better. 

The final paragraph [Lines 483-490] is rather confused and should be redrafted. Essentially, as it's about criteria for the recruitment of tutors for blended learning courses, the topic sentence should be something like: "The findings suggest a number of criteria for the recruitment of tutors for blended learning courses."

Response: this has been changed (lines 509-510), thank you – it now reads better. 

OVERALL

This study focuses on a small group of Education tutors (some part-time) and adult part-time learners. However, while not typical of mainstream undergraduate HE practice it may still be significant in providing evidence that challenges upbeat nostrums around the benefits of blended learning. In this respect it is also original. In terms of rigour, it is lacking (see comments on FINDINGS).

The study does deserves follow-up work: perhaps with a range of tutors in a range of academic disciplines in which blended learning is a significant part of course delivery (added to conclusion lines 520-522). Any follow-up studies should also employ richer evidence of educational effectiveness than student questionnaires and tutor interviews (for example, student interviews and/or focus group discussions) - (added to conclusion lines 522-526).

Reviewer 2 Report

The article presents all the necessary quality elements to be published in the magazine. No modification needed

Author Response

Thank you for taking the time to review this article, it is much appreciated.